# New Dental Implant with 3D Shock Absorbers and Tooth-Like Mobility—Prototype Development, Finite Element Analysis (FEA), and Mechanical Testing

**DOI:** 10.3390/ma12203444

**Published:** 2019-10-21

**Authors:** Avram Manea, Grigore Baciut, Mihaela Baciut, Dumitru Pop, Dan Sorin Comsa, Ovidiu Buiga, Veronica Trombitas, Horatiu Colosi, Ileana Mitre, Roxana Bordea, Marius Manole, Manuela Lenghel, Simion Bran, Florin Onisor

**Affiliations:** 1Department of Cranio-Maxillofacial Surgery and Radiology, Faculty of Dental Medicine, University of Medicine and Pharmacy ‘Iuliu Hatieganu’, 400012 Cluj-Napoca, Romania; avram.manea@umfcluj.ro (A.M.); gbaciut@umfcluj.ro (G.B.); veronicatrombitas@gmail.com (V.T.); ilmitre@yahoo.com (I.M.); florin.onisor@umfcluj.ro (F.O.); 2Department of Oral Rehabilitation, Faculty of Dental Medicine, University of Medicine and Pharmacy ‘Iuliu Hatieganu’, 400012 Cluj-Napoca, Romania; mbaciut@yahoo.com (M.B.); roxana.bordea@ymail.com (R.B.); 3Department of Mechanical Systems Engineering, Faculty of Machine Building, Technical University of Cluj-Napoca, 400114 Cluj-Napoca, Romania; pdcluj@gmail.com (D.P.); ovidiu.buiga@omt.utcluj.ro (O.B.); 4Department of Manufacturing Engineering, Technical University of Cluj-Napoca, 400114 Cluj-Napoca, Romania; dscomsa@tcm.utcluj.ro; 5Department of Medical Education, Faculty of Medicine, University of Medicine and Pharmacy ‘Iuliu Hatieganu’, 400012 Cluj-Napoca, Romania; hcolosi@umfcluj.ro; 6Department of Prosthetics and Dental Materials, Faculty of Dental Medicine, University of Medicine and Pharmacy ‘Iuliu Hatieganu’, 400012 Cluj-Napoca, Romania; mnole22@yahoo.com; 7Department of Surgical specialties, Faculty of Medicine, University of Medicine and Pharmacy ‘Iuliu Hatieganu’, 400012 Cluj-Napoca, Romania; lenghel.manuela@gmail.com

**Keywords:** dental implant, biomaterials, titanium, ISO 14801, fatigue test, finite element analysis

## Abstract

Background: Once inserted and osseointegrated, dental implants become ankylosed, which makes them immobile with respect to the alveolar bone. The present paper describes the development of a new and original implant design which replicates the 3D physiological mobility of natural teeth. The first phase of the test followed the resistance of the implant to mechanical stress as well as the behavior of the surrounding bone. Modifications to the design were made after the first set of results. In the second stage, mechanical tests in conjunction with finite element analysis were performed to test the improved implant design. Methods: In order to test the new concept, 6 titanium alloy (Ti6Al4V) implants were produced (milling). The implants were fitted into the dynamic testing device. The initial mobility was measured for each implant as well as their mobility after several test cycles. In the second stage, 10 implants with the modified design were produced. The testing protocol included mechanical testing and finite element analysis. Results: The initial testing protocol was applied almost entirely successfully. Premature fracturing of some implants and fitting blocks occurred and the testing protocol was readjusted. The issues in the initial test helped design the final testing protocol and the new implants with improved mechanical performance. Conclusion: The new prototype proved the efficiency of the concept. The initial tests pointed out the need for design improvement and the following tests validated the concept.

## 1. Introduction

Dental implants are widely used for oral rehabilitation of edentulous patients [1]. Besides the obvious advantages, dental implants have certain shortcomings, making them a constant research subject. The endosseous implants becomes ankylosed (functionally ankylosed) after osseointegration, without any periodontal ligament support [2,3,4]. Excessive occlusal loads (in relation to the bone-supporting capacity/implant bone interface) are together with peri-implant infections the most common cause of implant failure [5,6]. While in the early days of modern oral implantology osseointegration was the main concern, present day concepts focus more on improving the implant’s lifespan, reducing bone loss, and increasing patient comfort. Such goals can be attained by improving current implant designs to include shock-absorbing mechanisms (stress relievers/breakers) [4]. From a prosthetic point of view, it is not advised to use devices with mixed anchoring on both the implants and the natural teeth. Nonphysiological forces emerge which affect the bone. If, however, a prosthetic device is applied over an implant and a tooth, due to the forces that occur during the functioning of the dento-maxillary system, loosening of the prosthetic device can occur, as well as its fracturing or even the mobilization of the implant [7,8,9,10]. Obtaining implants that reduce force transfer from the superstructure to the surrounding bone has been a permanent subject of interest for researchers, leading to numerous studies and patents (US5425639A—Dental implant with shock absorbent cushioned interface 1994; US20090208904A1—Dental implant 2009; WO2015066438A1—Double-cushioned dental implant 2015). Work on this subject began in the 1980s with the IMZ implant system and has been going on ever since, including the previously mentioned patents and many others, but there is no currently available shock-absorbing implant for commercial use. This is mostly due to the complexity and frequent failure of these implants.

Natural teeth possess a certain degree of physiological mobility in the alveolar socket. This movement is mediated mostly by the periodontal tissues and it is influenced by several factors, such as: Tooth anatomy, alveolar bone anatomy, and patient age. Some authors (Parfitt, 1960) even mention changes in tooth mobility during the day (higher in the evening) [11]. The axial mobility of teeth with healthy periodontal support ranges from 0.01 to 0.03 (upper incisor) mm while the transversal mobility is situated between 0.05 and 0.2 mm for the same teeth group. These numbers decrease in the lateral regions, mostly due to root anatomy. Such measurements have been of interest as early as the 1950s (Mühlemann, 1951–1954; Parfitt, 1960) and have been studied ever since, underlining their importance in dentistry [11,12,13].

The negative effects of stress transfer from the implant to the surrounding bone and the potential counteracting of the phenomenon has been a constant topic of study in dentistry and in other medical fields such as orthopedics [14,15]. Stress breakers/relievers in the implant would greatly benefit the surrounding bone by decreasing the micro fractures due to overloading and the consecutive bacterial infiltration, decreasing bone loss, especially in the oral cavity. Other advantages in the case of dental implants might include: More prosthetic solutions, increased patient comfort, and an overall longer implant lifespan [14,16].

Another topic of interest in oral implantology concerns the microleakage at the implant– abutment interface/junction (IAJ). Its negative effect is visible through the consecutive crestal bone loss caused by the bacteria growing at the IAJ. Proper sealing of this space can provide a serious benefit [17]. Several studies were conducted in order to find solutions to evaluate reduce microleakage (Morse taper/cone–Morse implant connections proving to improve this issue, when compared to the classical external or internal hexagon connections) [18,19].

Implant screw loosening is a frequent occurrence causing increased patient discomfort and serious management issues for the dentist, especially in cement retained fixed restorations. Numerous studies have been conducted over the past years to determine the exact cause and effects of this phenomenon and also prevention and management solutions. Screw loosening also leads to increased microleakage at the IAJ and many potential mechanical complications [20,21,22,23].

Several published studies evaluate changes to the implant: Bone interface owed to the application of forces on the implant, depending on the implant external characteristics (macroscopic design, surface treatment) [24,25]. New biocompatible implant materials are also permanently being developed and tested for several applications [25,26,27,28]. Studies conducted in the past years on dental implants with modified designs (axial flexibility, internal shock absorbers) as compared to classic ones, are aimed to improve their lifespan and functionality [29,30,31,32]. 

Any new medical device requires extensive testing before entering mass production and clinical usage. The first tests always focus on the mechanical aspects of that certain device and indicate whether further studies can be conducted or redesigning of the concept is required. As far as dental implants are concerned, the mechanical tests refer to both static tests (bending, traction, compression, etc.) and also dynamic tests (fatigue tests). In vivo testing includes evaluating the implant’s osseointegration, surrounding soft tissue behavior, and many others, all involving great responsibilities and costs. This is why only after in vitro tests are performed, in vivo tests can be conducted. Time is one of the main factors to be taken into account in any sort of research; the shorter the time to obtaining relevant results, the better. Also, less time means decreased costs. The most common test used to evaluate dental implant mechanical properties (static and dynamic) is ISO 14801, dynamic loading test for endosseous dental implants. It was introduced in 2003, and updated in 2007 and then again in 2016 and will be replaced with “ISO/CD 14801” [33]. The dental implants can be tested in regular ambient conditions or in a saline bath at 37 °C. A total of minimum 11 implants are required in order to obtain the certification. One will undergo a static test, while the rest will be submitted to dynamic tests. While many argue that this test is incomplete and may sometimes generate just partially relevant results, it remains a valuable way to evaluate implants before entering production [34]. 

All these taken into consideration, the ideal testing method can be described as quick, relevant, reproducible, cheap, and safe. Unfortunately, they cannot be all found together, and several independent tests are needed to validate a certain concept. During the last two decades a different type of testing was implemented in this field, combining several advantages and presenting relatively few inconveniences. This technique is called “finite element analysis” (FEA) and represents a digital simulation of almost any available test. Although it has been used in engineering for more than 50 years, its medical applications are relatively recent and seem to provide valuable information when its limitations are understood. 

In dental medicine, simulations by FEA are quite common today. Its use is found in device testing and even tissue-response anticipation to certain devices or forces [35]. Studies have been conducted in general dentistry [36], endodontics [37,38], orthodontics, and especially oral implantology. Any implant test can be simulated (in vitro and in vivo), with almost immediate results and with a fraction of the cost of the physical tests. The sine qua non condition for obtaining relevant results for these simulations is the ability to exactly define all the parameters of the materials involved. Approximate values or lack of definition of certain parameters all together may result in an uncertain outcome. This means that most in vitro tests can be accurately simulated and can even provide information that is difficult to obtain from the classical physical testing, such as the behavior of the internal components of the implant, stress concentrations, and many more [39,40].

Despite the previously mentioned aspects, more and more studies are published including biological variables. In vivo studies are even harder to conduct because of the high risks, longer time, high costs, and numerous ethical aspects, making them sometimes impossible. Accurate digital simulations of such tests would be of immense value. Still, the aspect of defining the test’s parameters remains a big concern due to the complete lack of homogeneity of the tissues and the immense differences between patients [41,42]. Also, the tissues’ exact response is nearly impossible to predict and its remodeling in order to adapt to changing conditions is a parameter not taken into account by FEA [43]. A question arises then: Where should FEA no longer be used? The answer to this question is still debated. 

Mechanical testing for dental implants has been done for a long time now before entering mass production. Such tests include static and dynamic tests, aimed to simulating their functionality and durability while various forces are applied. The permanent upgrade of these standards shows a constant need of improvement and the desire to more accurately approach the real life conditions. 

Failing to pass these mechanical, in vitro tests makes any further in vivo testing useless. Even if they present the optimal biological properties, early mechanical failure will result in global implant failure. This is why it only makes sense to set the correct sequence of tests from the start. Even before the actual mechanical static and dynamic tests, their simulation by using FEA can reduce duration and costs and help reach a more favorable outcome. 

The aim of the present study is to propose and test a new implant design (patent pending a 2013 00275, OSIM, Romania) which can replicate the natural teeth mobility and reduce implant–bone interface stress. It benefits from a cushioning mechanism which is located between the implant and abutment, in order not to interfere with the osseointegration of the outer portion of the implant [3]. The whole concept was designed aiming to solve all the previously mentioned issues. The amortization mechanism inside the implant reduces occlusal forces passed on to the surrounding bone and also allows the abutment to have a natural tooth-like mobility. By filling the gap at the IAJ with the cushioning system, the present design will prevent microleakage. The specially designed screw and fixation mechanism of the abutment will remove any stress from the implant screw, making it not vulnerable to fracture or loosening. The present study describes the first tests ever conducted on this new implant concept and focuses mainly on the macroscopical aspect of the dental implant, as a preliminary phase in its development. It follows a reduced number of implants since its aim is to validate the implant concept and the test protocol design. The second part of the study describes the following tests, after improving the implant design and the testing protocol. In vitro tests were again conducted (dynamic testing according to ISO 14801) together with FEA, providing the expected results. It also describes the limitations of FEA and mechanical tests.

## 2. Materials and Methods 

In the first testing phase, six implants were fabricated (subtractive-milling) from Grade 5 titanium (Ti6Al4V). The implants were manufactured by BioMicron (Cluj-Napoca, Romania) following the authors’ instructions (3 of them presenting 3D shock absorbers: (P = Prototypes) P1, P2, and P3; and the other 3 had an identical exterior design (BioMicron ISC implant thread design) but the implant and the abutment were one piece: (S = Standard) S1, S2, and S3). The new design represented an original idea and required an initial in vitro testing phase, in order to identify and repair possible flaws as well as to evaluate its fatigue resistance. The exterior part of the designed implant was identical to the implant it was compared to, in order to rule out any additional variables. The only difference was inside the implant (between the implant and the abutment). The dimensions of the implant were: 4.5 mm in diameter and 13 mm in length. Other dimensions were also designed and will be produced and tested once the initial testing phase is passed and the required results are obtained. P1, P2, and P3 featured a body (the implant) and an abutment designed to provide mobility to the prosthetic element which was to be attached on the abutment (Figure 1).

The cushioning mechanism was placed between the implant and abutment. This was the only positioning that could provide the required amortization and mobility for the superstructure, without interfering with the exterior design of the implant and its osseointegration. For this phase of the test, the cushioning mechanism was made from latex (entire implant-abutment interface, sheet of 0.2 mm thickness) and Viton (between implant platform and abutment, in the specially designed grove of the abutment, intended to provide supplementary sealing, prefabricated O-ring with 4 mm outer diameter, 3 mm inner diameter) (Figure 1). Both latex and Viton are FDA (Food and Drug Administration) approved to be used in the food industry and in medicine but are not implantable. For this in vitro stage of the experiment, they were very well suited, providing the necessary resiliency and resistance, in order to observe the implant behavior under mechanical testing. In future in vivo testing, they will be replaced by medical grade silicones with appropriate rigidity and biological properties. The intended maximum mobility was 0.01 mm when applying a 30° force on the abutment, for forces situated in the physiological masticatory range (in this case 30–300N). S1, S2, and S3 had the same external shape as P1, P2, and P3 but did not possess any shock absorbers. They were made of a single titanium block (one piece). 

The abutment had an internal hexagon anti-rotational system and its axis movement was controlled by a specially designed mechanism, composed of 3 pins (located circumferentially at 120°) that were locked in their functional position by the implant screw. This design allowed for a perfect coronal displacement locking while still allowing the implant to have the 3D movement. Another very important advantage of this concept is the fact that there was no possibility for implant screw fracture. 

The abutments were covered with custom-made hemispherical caps. This particular shape was required to ensure a uniform force distribution to the entire abutment during the testing phase. Without it, the head of the testing machine would only apply the loads on the edge of the abutment, developing an irregular internal stress distribution which would lead to irrelevant results. The center of the hemisphere would be located at 11 mm above bone level (Figure 2). 

The tested implant concept was designed to be easy to use by the dentist. The body of the implant was inserted into the bone by using a simple hexagonal torque wrench and a standard insertion protocol. The abutment, that also included the cushioning mechanism, was then easily fitted inside the body of the implant. This component of the implant can be replaced at any moment during the life of the implant with ease. Several types and sizes of implants and abutments are available, according to the requirements of a certain case. 

The implants were inserted perpendicularly in a pig mandible (ex vivo tissue, exempt from Ethics Committee approval). The purpose of placing the implants in bone blocks was to evaluate the macroscopical changes that the implant induces to the supporting bone after undergoing a number of dynamic test cycles (in vitro simulation of masticatory cycles). These changes in implant stability were to be evaluated by using the Periotest. Three millimeters of the implant were left above bone level. Bone blocks measuring 15 mm in height, 15 mm in width, and a length of 10 mm, were sectioned and fitted into the dynamic testing device (Figure 3) (testing device, INSTRON 8870 (The National Institute of Research and Development in Mechatronics and Measurement Technique (INCDMTM), Bucharest, Romania) serial No. 8872K2654, actuator model A1740-3002 serial No.3113/07 with force cell serial nr. 58892, load capacity 10.000 N belonging to precision class 0.5% with an accuracy of ±0.25%) at a 30° angle, in order to comply with ISO 14801/2016 (Dynamic fatigue test for endosseous dental implants) standard [33]. 

Initial mobility was measured for each implant separately using the Periotest Classic (Medizintechnik Gulden, Germany), at the level of the 3 mm portion of the body of the implant, located above bone level. A telescopic arm was created in order to offer the Periotest positions which can be replicated for each measurement, and was attached to the testing device. The head of the measuring device was placed perpendicularly on the implant axis (±15°), in position 1 (0°), position 2 (90°), and position 3 (180°), and measurements were repeated three times for each stage (Figure 4). The initial Periotest values were documented in order to compare them to the measurements obtained in the next stages. The Periotest is a commonly used device in oral implantology designed to measure the initial stability of dental implants, describing the implant mobility relative to the surrounding tissue (bone). The Periotest functions by emitting transient vibrations and then measuring the reaction of the peri-implant tissues, offering a quantifiable measure unit. It can be used in all the implantological treatment stages, not only after insertion but also in the healing period to assess osseointegration and after the prosthetic treatment is finished, throughout the implant lifespan to detect early signs of peri-implantitis.

A number of 1000, 10,000, 100,000, and 350,000 testing cycles (moving in a sinusoidal wave from 30 to 300 Newton (N) with a frequency of 15.0 Hz) were applied to each implant and Periotest measurements were repeated for the implants after each cycle set. The testing was conducted in a 22 °C temperature environment. Testing was not conducted in liquid media. The ISO standard stipulates that such tests can also be conducted in a saline bath at 37 °C in order to be closer to the environment found in the oral cavity. “For endosseous dental implants that include materials in which corrosion fatigue has been reported or is expected to occur, or for systems that include polymeric components, testing shall be carried out in normal saline or in physiologic medium. The fluid and the test specimen shall be kept at 37 °C ± 2 °C during the testing. For all other systems, testing may be conducted in air at 20 °C ± 5 °C. The testing environment shall be justified and reported.” (ISO 14801 Standard). For this phase of the test, the dry conditions were chosen in order to eliminate any supplementary variables when comparing the different implant types and to be able to perform the Periotest measurements without removing the blocks containing the implants from the test machine’s vise (and tank containing the mentioned liquid, if present). Another reason for not conducting the test in liquid media in this stage was the maximum frequency of 2 Hz allowed in case of testing the implants while immersed. 

Any possible mobility differences which might have occurred between experimental implants and control implants or between the same type of implants before and after every test cycle was registered respectively. Comparisons were provided between the means of the measured degrees of mobility by applying statistical nonparametric tests (the U test, the Mann-Whitney, and the Wilcoxon test for paired samples, respectively).

The next step of the project consisted in manufacturing ten implants (Grade 5 titanium alloy (Ti6Al4V) by using a subtractive technique (milling), at the BioMicron implant manufacturer in Cluj-Napoca, Romania, according to the indications provided by the authors (Figure 5). The intended mobility of the abutment in relation to the implant was of up to 0.1 mm transversal and 0.025 mm axial, for forces higher than 30 N. The concept was tested and proved valid by our previous tests. The implants’ length was 13 mm with a 4.5 mm diameter. The outer part of the implant had no thread, as in the digital testing, in order to remove any irrelevant variables and to get the best comparison between the results. The cushioning mechanism was fabricated from latex and Viton. This second design benefits from an interior design that eliminates the areas of minimal resistance found in the first test and also an improved implant screw.

Before the mechanical tests were conducted, a digital simulation by FEA was performed at the Technical University of Cluj-Napoca. The outer design of the implant was modified in comparison to the initial design by removing the thread, both for the digital and the physical tests that followed, in order to eliminate as many variables as possible. This did not interact with the outcome due to the fact that the test focused on the internal components and behavior of the implant. Since the static tests (applying a linear increasing force until the point of implant failure) were already showing good results from the previous tests, the new study only focused on the dynamic testing to evaluate fatigue resistance. FEA was conducted using SolidWorks dedicated simulation module.

The digital analysis started by breaking the implant into 81,234 tetrahedral finite elements, forming 127,612 nodes. For all the three-dimensional model’s components, an elastic isotopic constitutive model was adopted (Figure 6a). 

All the implant’s metal components were made by titanium alloy (Ti6Al4V), with the following properties: Longitudinal elastic modulus (Young): 114,000 MPa;Ratio of transverse contraction (Poisson): 0.3;Yield strength: 860 MPa;Tensile strength: 930 MPa.

The force-absorbing mechanism situated between the implant and abutment was made by latex (will be replaced by medical silicones for the in vivo testing) with the following parameters:Longitudinal elastic modulus (Young): 0.5 MPa;Ratio of transverse contraction (Poisson): 0.45;Yield strength: 20 MPa.

The last component, the sealing gasket situated between the abutment and implant as well, designed to prevent any infiltration at that level, was made by Viton (will be replaced by medical silicones for the in vivo testing). It had the following parameters:Longitudinal elastic modulus (Young): 8.56 MPa;Ratio of transverse contraction (Poisson): 0.48;Yield strength: 16.2 MPa.

The implant’s resistance was evaluated by static analysis, reproducing the exact parameters of the physical testing:Complete immobilization of the exterior part of the implant (only 3 mm of the coronal end of the implant left unlocked) (Figure 6b);Adherent contact between the implant’s components (Figure 6c);300 N forces applied at a 30° angle on the abutment of the implant (Figure 6d).

Dynamic testing was simulated by applying forces ranging from 30 N to 300 N in a sinusoidal manner with a 15 Hz frequency, exactly like in the physical testing. Static testing was also simulated with FEA, under the same conditions as described by ISO14801, and by rotating the implant around its axis by 10° for each measurement to assess the variations in maximum internal stress dependent on implant rotation. No significant variations were expected to be found. 

According to ISO 14801, for the fatigue test, dental implants were positioned in a less than ideal setting (3 mm above “bone level” at a 30° angle). The abutment was covered by a hemispherical cap, specially designed for this test, to ensure a uniform force distribution on the abutment, not only on its edge (Figure 7). A total of minimum 11 implants were required in order to obtain the certification from which 1 will undergo a static test, while the rest will be submitted to dynamic tests. 

Static testing required the implant to be fitted into the machine as mentioned before and a constantly increasing force to be applied on it until failure (Figure 8). The static tests were not repeated since our initial results were very good, and the 612 N failing force was considered for this test as well. 

Dynamic testing (Figure 9) followed, with the previously mentioned forces, using a sinusoidal pattern (forces ranging from 10% to 100% of the dynamic test force value, 30 to 300 N for the present test) with a 15 Hz frequency. In order for the implant to pass the test, a lifetime of 5 million test cycles should be achieved. The tests were conducted on the same Instron testing machine (Model 8870, series no. 8872K2654, actuator model A1740-3002, series no. 3113/07, with the force cell series 58892, load capacity of 10.000 N, precision class 0.5% with ± 0.25% accuracy). The measurements were carried out with digital calipers. 

The second testing phase was carried out in the Biomechatronics laboratory at the National Institute of Research and Development in Mechatronics and Measurement Technique in Bucharest, Romania, between January and February of 2018. 

## 3. Results

During the first test session, the P2 and P3 implants failed (one or more components suffered macroscopical damage, abutment in this case, making the entire assembly not function as intended) after a number of 10,000 and 35,045 cycles, respectively. (Table 1, Figure 10a). Both of the test samples failed at the point where the abutment entered the implant body (at the locking pins level), an area that proved to be the weakest point of the implant concept. The abutment had less than 0.5 mm width at that certain point. All the other implant components worked as designed, without any damage or unwanted displacement until the point of implant faiulre.

An obvious difference in displacement was visible on the computer generated graphic (Figure 11) between the one-piece implant (S1) and the prototypes (P2 and P3). While the displacement for S1 ranged in a 0.01-mm interval, the displacement for the implant prototypes was situated between 0.03 and 0.06 mm for forces higher than 30 N. 

The bone block into which P1 was inserted fractured after it was fitted into the testing device (Figure 10b), due to increased momentum used when fixating the bone block in the test machine’s vise. 

Three Periotest measurements were conducted at the designated locations for each implant in each phase but they were inconsistent even at the intial measurements (before applying any dynamic loads, differences of up to 10 Periotest units), even if the measuring device was anchored in a specially designed arm to eliminate user-dependent variations. The Periotest measurements after 1000 and 10,000 test cycles showed the same issue. 

Although the initial testing protocol could only be carried out partially, it managed to prove the concept. The displacement of the abutment in relation to the implant body proved to be considerably increased in comparison with the standard (one-piece) implant. Its values were situated in the expected range.

Testing then continued to verify the validity of the implant concept. Implant P1 was tested statically, mounted in the same 30° angulation, with 3 mm of the implant being left above the test machine vise (not inserted in any bone block). (Testing device: Hounsfield, H10KT, with force cell No. 0198107, load capacity of 10.000 N), failing at 548 N and resisting up to 612 N (Figure 12). Testing for S1 was carried out as planned. S2 implant was also tested using the same protocol as P1 to compare results (Table 2).

The results for the second test stage showed improved results. During FEA, static testing was conducted in several positions of the implant (rotating the implant every 10° around its axis) and the variations in the implant’s components behavior was obvious. The results are mentioned in Table 3. These values are calculated at a portion of the abutment corresponding to the neck of the implant (Figure 13), the area of minimal resistances suggested by all our previous testing.

In this table, α: Angle that defines the orientation of the implant in relation to yOz plane; σ_ech,max_: Maximum von Mises stress during static analysis.

The difference from 298.5 to 374.1 MPa in the maximum stress represents a 25.3% increase, provided just by the difference in rotation of the implant, clinically obtained during the placement (screwing) of the implant in the alveolar bone. These stresses are calculated at a 300 N implant load. Increasing the implant loading will obviously increase the stress, resulting in large differences just due to the rotation of the implant.

The 30° rotation was considered for the following dynamic tests (fatigue), since it represented the worst case scenario, developing the highest stress values (374.1 MPa). The Wöhler curves (fatigue resistance) for each material (titanium alloy, latex, and Viton) are presented in Figure 14. 

Comparing the information in Figure 14 and Figure 15 we can observe that the maximum von Mises stress developed under regular loading conditions (300 N) upon the implant and abutment is under the Wöhler curve for Grade 5 Titanium. This translates into the fact that the implant can work under such conditions for an unlimited period of time, a theory confirmed by the fatigue test results which suggest a lifespan of more than 10 million loading cycles (Figure 15). 

The physical fatigue test was performed for three of the ten implants until this point. The rest are scheduled for testing in the following months. Even at this point, the results are very promising, since they are very similar to those of FEA. The first implant resisted at all the 5 million test cycles, without sustaining any damage. The testing stopped since the target was reached. The second implant test was stopped due to a power shortage which allowed the test arm of the machine to collapse on the cap of the implant, fracturing it, just after reaching 500,000 test cycles (Figure 16). Prior to that moment, the implant was not showing any signs of wear. For the third implant, faulty assembly was simulated, resulting in failure after 266,103 cycles. The early fracture was attributed to the incorrect position of the implant components. Implant No. 3 (Figure 5) did not have the screw fully tightened which translated into incomplete locking of the pins. The damage and the incorrect position of the screw are visible in Figure 17.

The rest of the fatigue tests will be carried out soon and similar results are expected. If all test parameters involving human error (faulty assembly) and machine error (power shortage) are removed, the “5 million” milestone should be reached for all the prototypes. Figure 18 shows a comparison between mechanical testing and FEA, with almost identical results, showing the same point of minimal resistance (implant No. 2). The implant would not have failed at 500,000 cycles under normal testing conditions.

## 4. Discussion

Although the expected results were not achieved, as far as resistance to dynamic testing is concerned, in the first test phase, the static test had satisfying results. The expected mobility was reached in comparison to the standard (Figure 11) [45], proving the concept. As mentioned before, by achieving this mobility, the currently tested implants have the potential of being splinted with teeth, which is not recommended with other implants [2,8]. This aspect must be proven by further testing of the present implant. In a combined teeth-implant supported bridge, when forces are applied at the tooth level, the implant tends to tilt towards the tooth. A zero-mobility implant or with just axial mobility could not deal with such lateral forces and would transmit them to the peri-implant bone in a nonphysiological manner. The three dimensional movement and force absorbance will spare the bone of these forces. Although bridges between a tooth and an implant which are close to each other have a good success rate, larger fixed restorations are to be avoided. With implants that have a mechanical behavior close to natural teeth, even full arch bridges can be used. Some recent studies show satisfying results for combined implant and tooth support restorations. But, according to them, this is only true in certain cases, when several requirements are checked. Not all clinical situations benefit from the same satisfying outcome which limits the treatment options [46]. The aspect of splinting teeth and dental implants still remains a controversial and much debated topic.

Other studies publish data on implant failure under dynamic testing for similar dimension implants (4.1 mm × 12 mm), with a classical implant–abutment setting. Testing was conducted using ISO 14801 as a guide but the protocol was customized. Some implants were tested in ambient conditions while others were tested in saline solution. The frequency was of 2 and 30 Hz, respectively, (not 15 Hz as imposed by the standard). The implants were not fixated into the test machine’s vise but in bone block analogs. Forces ranging in a sinusoidal manner from 20 to 420 N were used. The implants failed from 79,926 to 3,918,266 cycles [47]. The currently tested implant prototype showed lower failure values but the results are encouraging since it was the first ever mechanical test and it presented a more complex design. 

After analyzing the results and the fractured implants, further improvements to the implants and testing protocols were made. The designs of the internal part of the implant and the abutment were changed. The latex and Viton used as shock absorbers and sealing systems, respectively, will be replaced with implantable materials with similar mechanical properties in the future for in vivo testing since they provided the expected results. When analyzing the implant components individually, they proved to work as designed. The implant screw did not sustain any damage until total implant failure. The locking pins worked as they should, not allowing the abutment to be macroscopically displaced in a coronal direction, even after several test cycles. The cushioning mechanism did not sustain any macroscopic damage but its presumed efficiency must be further tested. Also, its design can be modified in order to provide a more similar function to the ligamentous structure of the periodontal structures. From a microleakage point of view, the implant concept has to undergo more tests to compare it to the already available implant–abutment interfaces’ behavior. Since the previously published articles on the topic of microleakage find the Morse taper implant connection to show less infiltration between the abutment and implant (still not a perfect seal) [18,19], the currently tested implant will be compared to this connection type to evaluate the potential improvement. Cement retained crowns applied on the two-piece implants seemed to provoke less microleakage between implant and abutment compared to screw retained crowns (40 vs. 60 µm) after several years of implant function [48]. This might be due to several factors such as the partial force absorbance caused by the cement’s elasticity and the ability of the cement to fill the voids between implant and abutment. Because of the decreased microleakage in the cement retained restorations, the new implant concept will be tested with these types of crowns and observed to see if it improves the currently available implants’ performance in terms of bacterial infiltration between its components.

Another innovative way of evaluating potentially present bacteria between implant and abutment (implant and healing screw) and their effect on the surrounding tissues is by employing volatile organic compounds (VOC) analysis [49]. Although it is not a commonly used test, it can provide valuable insight in bacterial activity (not only in bacterial presence). Future testing of the present implant concept will include VOC analysis to evaluate potential bacteria in the cushioning mechanism (in vitro testing) and also host response to the implant (inflammation in the peri-implant tissue if present) in conjunction with VOC in the in vivo phase of the test.

The testing protocol will be changed as well, due to limitations observed in the measurements conducted with the Periotest (inconsistent results). Similar studies have shown that Periotest values might differ depending on examiner and angle, which might be a source of error for our tests [50].

Similar studies tried to achieve tooth-like mobility and shock absorbance, but only in an axial direction. Pektaş and Tönük [29] pointed out the problem of high stresses within the implant system and in the jawbone in mixed support bridges. Their solution consists of a vertical shock-absorbing mechanism in the abutment of the implant. Although this proved to be useful for pure axial forces (which are rarely applied on teeth or implants) [51], it is not efficient with lateral forces. In 2001, Gaggl and Schultes [52] state the importance of shock absorbing in dental implants and use a silicone ring between the abutment and implant. They concluded by indicating such implants in mixed support bridges and finding a lower degree of bone loss due to overload. No information was, however, given about the lateral forces. 

All of them obtained good results, but the periodontal system is a 3D structure which allows the tooth to move in all spatial directions, not just vertically. From the shock-absorbing point of view, in an axial direction, most forces will be transmitted to the apex of the implant and to the apical portion thread. In 3D shock absorption, forces can be transmitted in a more physiological manner to the surrounding bone. Also, in splinting implants with no mobility or only axial mobility and adjacent teeth which possess 3D mobility, various issues may arise. Such implants do not present mobility and shock absorbance in a mesiodistal and orofacial direction [53].

The results of the second test phase provided valuable insight in implant design, development, testing, and function. First of all, it restated the utility of FEA combined with physical testing of the dental implants. There were obvious similarities between the results of both tests but also important differences that must be discussed.

Other studies suggest using this method to replace physical testing [54] in the early phases of implant development, in order to evaluate fatigue resistance, with reduced costs and time. Physical testing in conjunction with FEA are also recommended by some authors, proving the similitude of their results during implant [55] and even prosthetic restorations testing [56]. 

One of the main advantages in FEA concerns the time-related aspect. In our experience, producing the implant and testing took up to 4 days/sample (1 day for production, assembly, and inspection, and up to 3 days for the fatigue tests). FEA only took a few minutes to introduce all the parameters into the system and seconds to run all the calculations before receiving the results. Test costs are also important in any study. While producing and testing a set of implants can generate extremely high costs, FEA only cost a fraction of that. 

During the FEA conducted for this study, an additional problem was observed: The differences in stress developed within the implant’s components depending on the implant rotation. This is especially obvious for implants with particular designs such as this one. The most favorable position provided decreased stresses by 25.3% compared to the worst case scenario and a 17.54% stress decrease compared to average values. None of the currently available dental implants is perfectly round (interior and exterior) which means that such variations exist in all implants. 

Markings can be made on the neck of the implant to allow the practitioner to obtain the optimal rotation during insertion. These markings can have correspondents on the hand wrench to eliminate the need of removing it to verify the exact position of the implant. For this design, the optimal rotation was found to be 190°. Close rotational values, such as 180° or 200°, also provided favorable stress values (308 and 300 MPa, respectively, the second and third lowest values obtained) which translated into some tolerance regarding rotation during insertion. The relative error between 190°and 180° is: $\varepsilon =\frac{308-298.5}{298.5}\times 100\%=3.18\%$. For a thread pitch of 0.8 mm, considered among the most favorable in terms of stress distribution [57,58], a 10° rotation translated into a 0.0222 mm advancement. At 200°, the implant will be placed 0.0222 mm more apically while at 180° it will be 0.0222 mm coronary compared to the 190° rotation. $z=\frac{0.8\times 10}{360}=0.0222\;mm=22.2\;\mu m$. For a 1.2-mm thread, the same rotation translated into a 0.0333-mm difference in insertion depth. This means that optimal rotations can be obtained with little effect on the implant position. 

A cyclic variation in the stress levels can be observed, due to the hexagonal shape of the implant’s interior. Very little published literature regarding this topic is available, although its practical implications can be of crucial importance. This aspect regarding implant rotation would require at least 35 implants to test, translating into months of testing with extreme costs and many sources for error that might seriously influence the results. Also, it would only generate observational results without the ability to calculate the exact stresses developed inside the implant. In this case, FEA proved to be an extremely useful test method. To further test the theory regarding the implant rotation and consolidate all our findings during FEA, mesh refinement studies and sensitivity analysis of material parameters must be conducted. Such analyses will provide even more accurate and close to physical testing results. 

The similitudes between the results obtained in both fatigue test types are obvious (Figure 18). The areas of minimal resistance/maximal stress found in the digital simulations coincide with the points of failure from the mechanical testing (independent of the cause of failure). The association of these tests helped us obtain the present results (failure due to fatigue after 10,000 to 35,000 cycles in the early stages, reaching the 5 million cycles mark in the present). Since modifying the prototype design and mechanically testing it each time would have been impossible, FEA was used for every modification made to the original design until the expected results were obtained, leading to the production and testing of the improved version. The differences between the digital and mechanic tests are as obvious as their similarities. This was caused mainly by the difficulty in defining all the parameters required in FEA, even if we were only referring to materials with constant, linear parameters. Other variations might be due to human error during physical testing (manufacturing, assembly, testing) and even machine failure during any of these stages. Although the process of producing the dental implants today is extremely precise and the practitioners’ experience is constantly improving, errors might be also present in clinical situations. FEA on the other hand, provides perfectly correct results which might seem to be an advantage but it cannot anticipate the errors that can occur in everyday practice.

A high number of studies present the use of FEA in oral implantology. They evaluate biomechanical aspects [59,60,61,62,63] but also tissue-related issues [64,65,66,67,68]. Such tests might provide only partially relevant information since the accurate definition of tissue parameters is impossible. Also, unlike inert materials, tissues react in ways that cannot be anticipated by even the most sophisticated algorithm. Of course, such results would be very valuable since removing the implant and other tissues from the human body for scientific purposes is impossible, but much attention should be paid when interpreting any findings. Recent studies go into extreme detail even regarding the tissues [68] by defining the areas of interest as heterogeneous bodies, separating the cortical bone from the internal midollar bone, and even dividing the studied object (human mandible, in this case) into several areas with different mechanical properties according to the general population values. Such studies are the closest simulations to in vivo conditions that can be done by FEA and provide valuable information. 

While the initial phases of implant development are possible with the help of FEA (validated by numerous studies, including our present research), the following steps should be approached differently. The lack of validation for the usage of FEA in studies that involve human tissues is also mentioned in literature review studies [41,69].

## 5. Conclusions

Within the limitations of the present study, the conducted tests prove the efficiency of the prototype. Manufacturing dental implants and testing them by using classical means and also digital simulations helped improve the concept from the early stages until obtaining a functional prototype. The differences in implant mechanical properties were immense in the final tests compared to the initial ones. 

After the first set of results, the project focused on improving the fatigue resistance of the implant in order to comply with the ISO 14801 standard. The intended 3D mobility was obtained but must be further tested and calibrated to get exact and 100% replicable results. The cushioning effect must be evaluated in vitro and in vivo afterwards. The microleakage at the IAJ must still be observed. The behavior of mixed support prosthetic restorations must be evaluated in vitro (mechanical testing and FEA) and especially in vivo afterwards. 

The present study also analyzed the pros and cons of the digital testing methods in Oral Implantology and touched less-studied aspects, such as the considerable differences in maximum stress developed in the implant depending on rotation. Also, the importance of FEA and its limits are accurately depicted, validating its usage in the in vitro stages of implant development. In following (in vivo studies), if digital simulation methods are chosen to be used, their results must be interpreted with caution. FEA is a valuable tool in implant development and provides information impossible to obtain using classical testing methods. Nevertheless, it can only be used as an adjuvant to already validated methods and can never fully replace any of the established testing phases.

In conclusion, although some variables are yet to be eliminated and certain improvements are required, the present implant concept proved to be functional and represents a topic worth further research. Many more implant prototypes need to be manufactured and tested (in vitro and in vivo) before reaching a finite product.

## 6. Patents

Patent pending, a 2013 00275, OSIM, Romania, “Implant dentar care reproduce mobilitatea fiziologica a dintilor naturali” (Dental implant with natural tooth like mobility).

## Figures and Tables

**Figure 1 materials-12-03444-f001:**
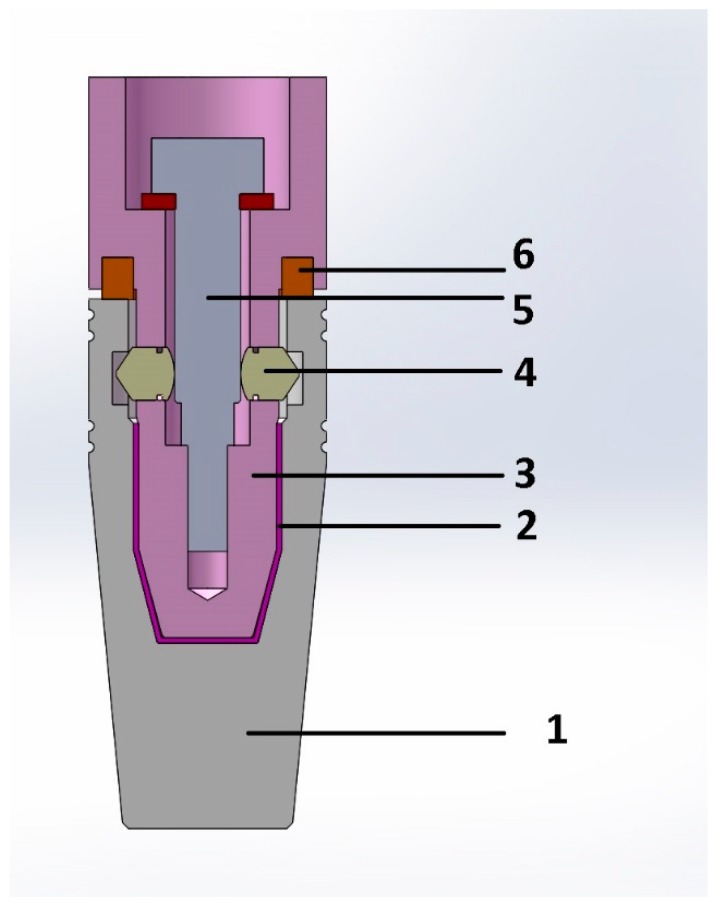
Initial drawing of the implant concept (SolidWorks): (**1**) Implant body, (**2**) cushioning mechanism, (**3**) abutment, (**4**) locking pins, (**5**) implant screw, (**6**) O-ring (cushioning mechanism).

**Figure 2 materials-12-03444-f002:**
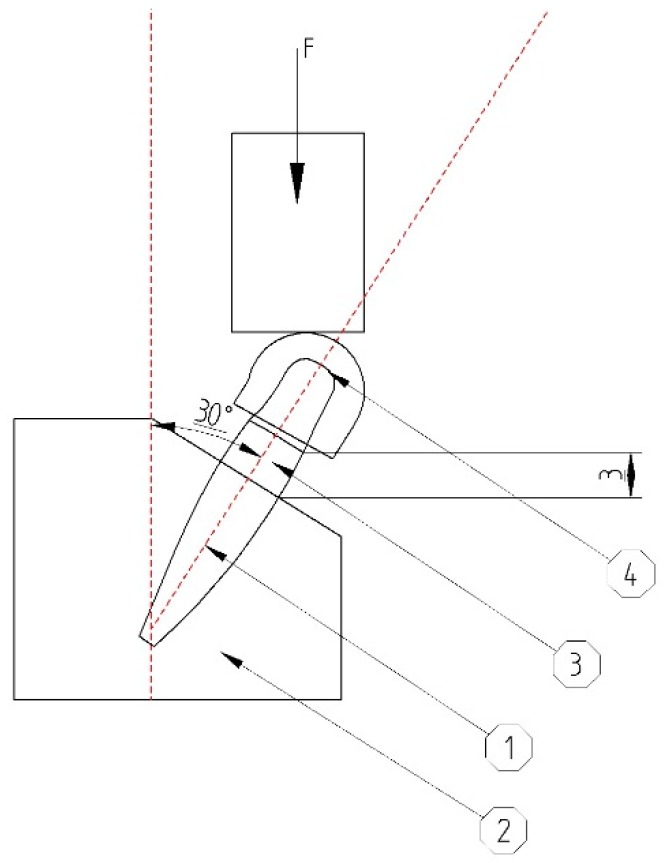
Scheme for mounting of the implant in the testing machine according to ISO 14801. The body of the implant (**1**) is mounted in a block, (**2**) bone in our case, with its coronal portion (**3**) 3 mm above the bone level. It is angled at 30° in order to stimulate less-than-optimal placing of the implant. The force is applied on a hemispherical cap covering the abutment (**4**) specially designed for this test.

**Figure 3 materials-12-03444-f003:**
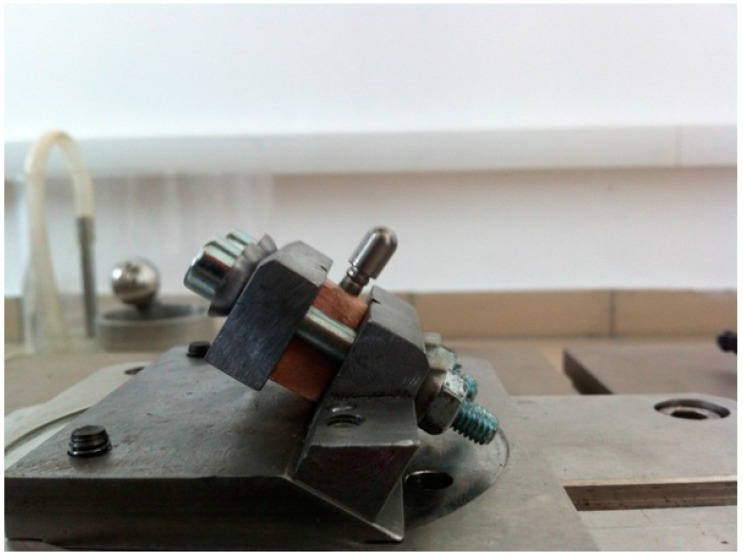
Mounting of the bone block containing the implant in the testing machine (INSTRON 8870 [44] at The National Institute of Research and Development in Mechatronics and Measurement Technique, Bucharest, Romania), at a 30° angle, with 3 mm of the coronal portion above the bone. The abutment is covered by a hemispherical cap specially designed for this test.

**Figure 4 materials-12-03444-f004:**
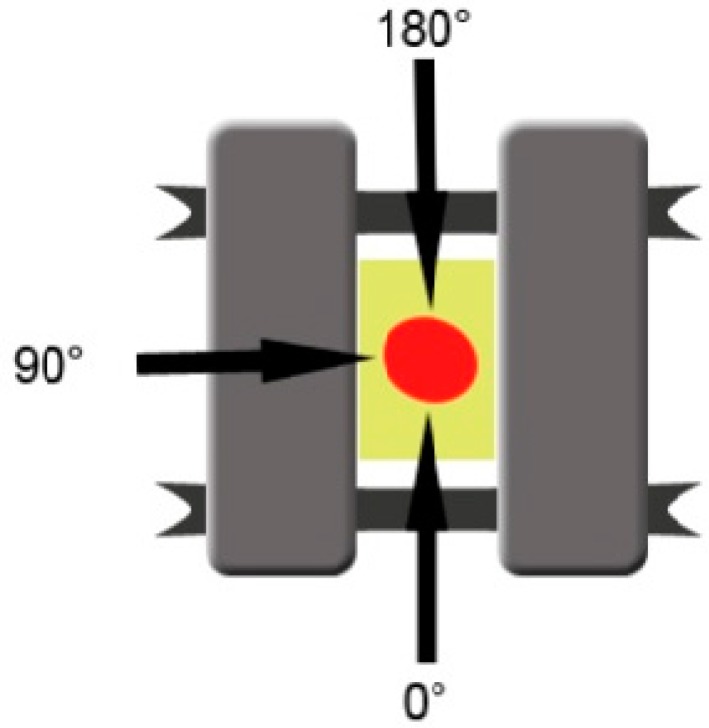
Points indicating the directions of the Periotest used in this study to examine differences in mobility of the implants in various stages of testing. The 270° angle could not be used for measurements because of the arm of the testing machine.

**Figure 5 materials-12-03444-f005:**
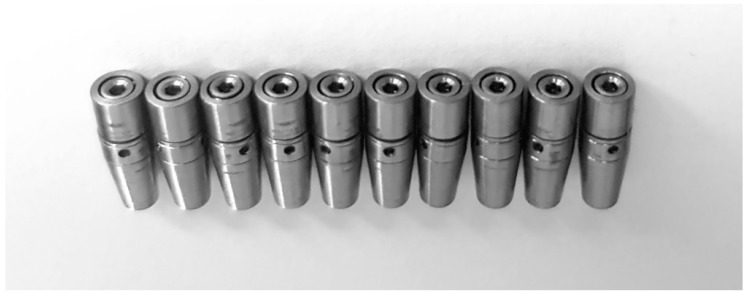
The entire batch of implants before testing. Implants numbered 1 to 10 from left to right.

**Figure 6 materials-12-03444-f006:**
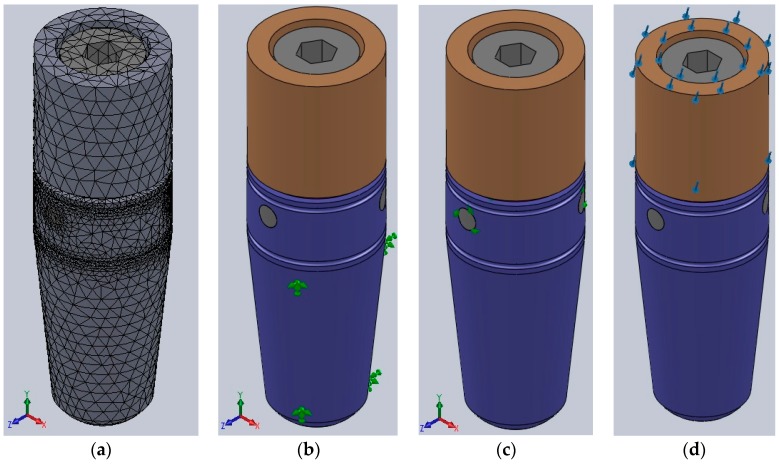
(**a**) Implant’s 3D model discretization into finite elements; (**b**) blocking of the outer surface of the implant; (**c**) adherent contact between the implant’s components; and (**d**) force vectors.

**Figure 7 materials-12-03444-f007:**
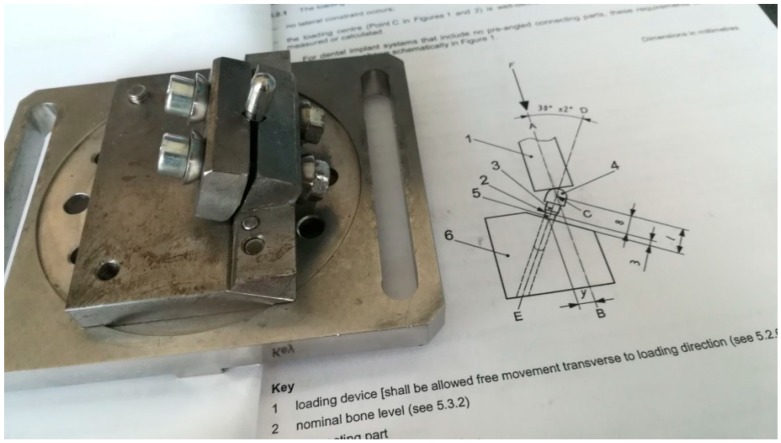
Implant positioned for the second round of mechanical (dynamic) testing. This protocol eliminated the bone blocks present in the first tests. The implants were fitted directly into the testing machine’s vise.

**Figure 8 materials-12-03444-f008:**
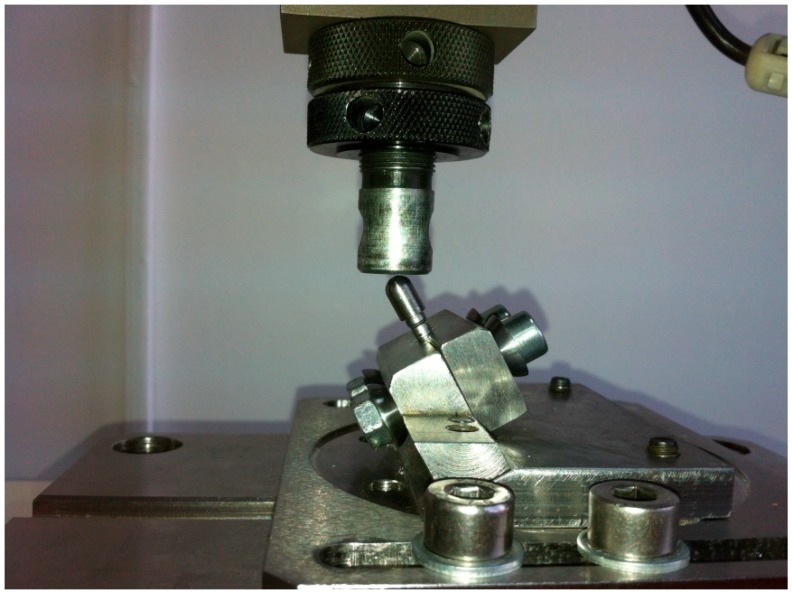
Commencement of the static test.

**Figure 9 materials-12-03444-f009:**
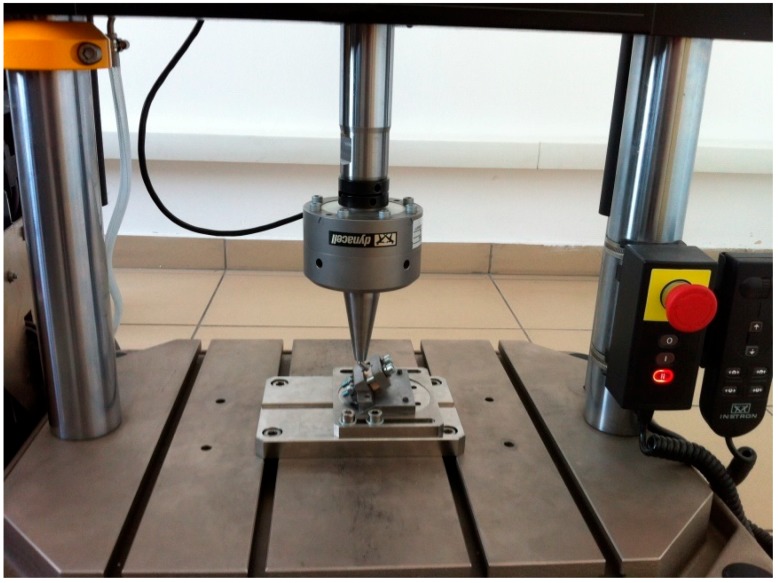
Dynamic implant testing (second implant set).

**Figure 10 materials-12-03444-f010:**
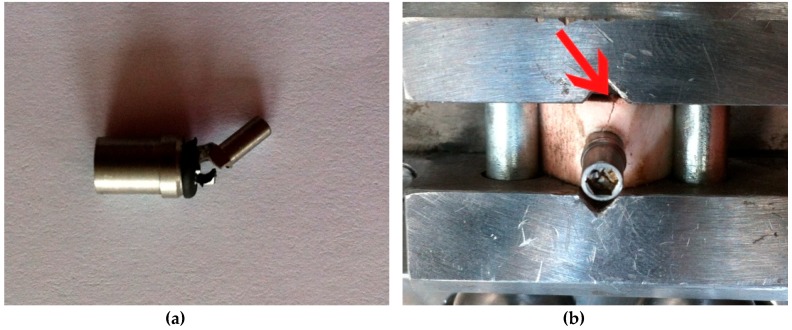
(**a**) Abutment fracture in P2 implant; (**b**) fissure of the bone around P1 implant.

**Figure 11 materials-12-03444-f011:**
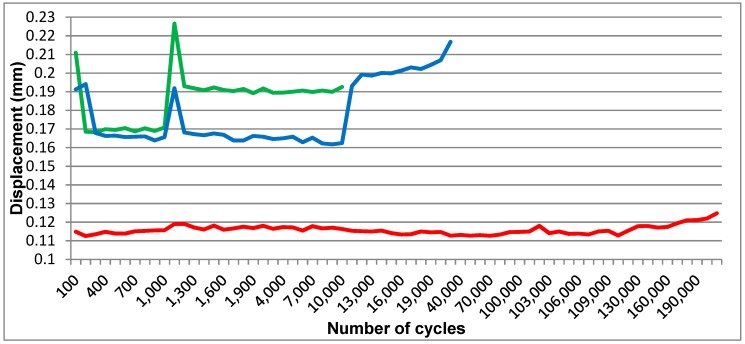
Dynamic testing for S1 (red), P2 (green), and P3 (blue). The peaks present in the green and blue line symoblize the removal and then aplication of the test device load cell in order to perform the Periotest measurements.

**Figure 12 materials-12-03444-f012:**
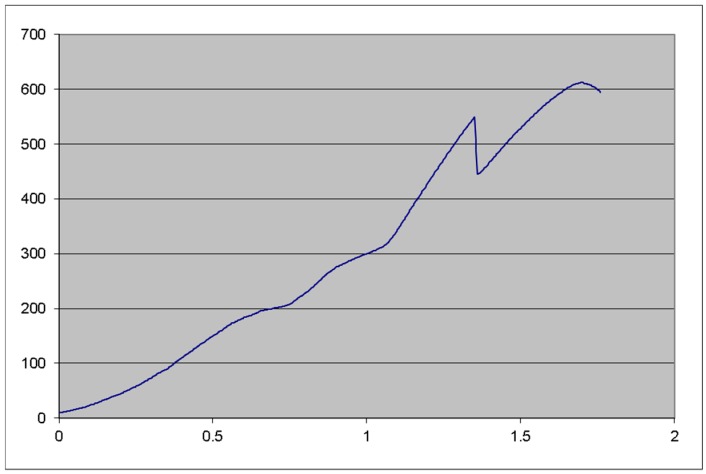
Statical testing of P1 implant. There is almost a linear relation between the force applied and the displacement. The implant suffered a small fracture at 548 N but it failed entirely only at 612 N. (*X* axis, displacement mm; *Y* axis, force, N).

**Figure 13 materials-12-03444-f013:**
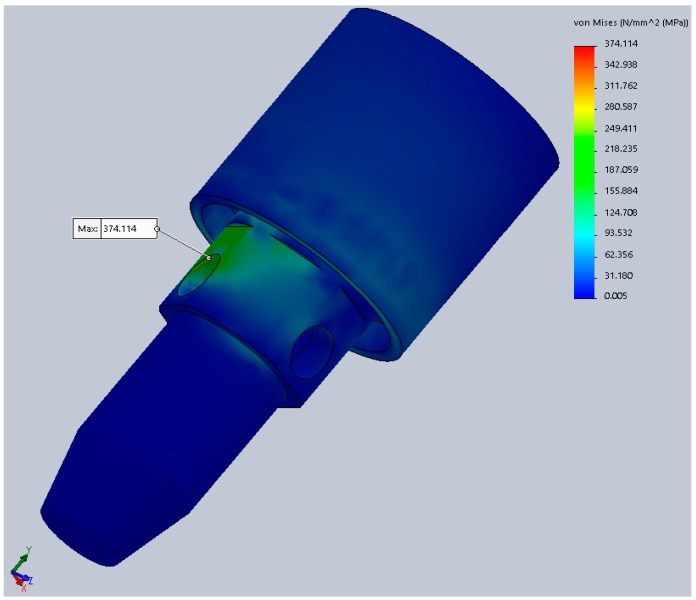
Maximum von Mises stress during static analysis (portion of the abutment corresponding to the neck of the implant).

**Figure 14 materials-12-03444-f014:**
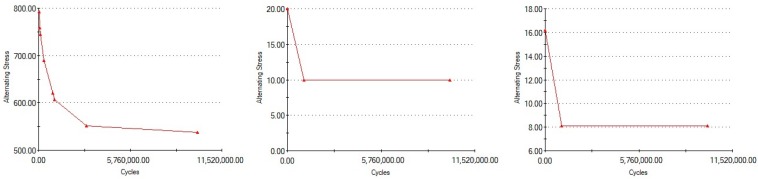
Wöhler curves for Grade 5 titanium (left), latex (center), Viton (right).

**Figure 15 materials-12-03444-f015:**
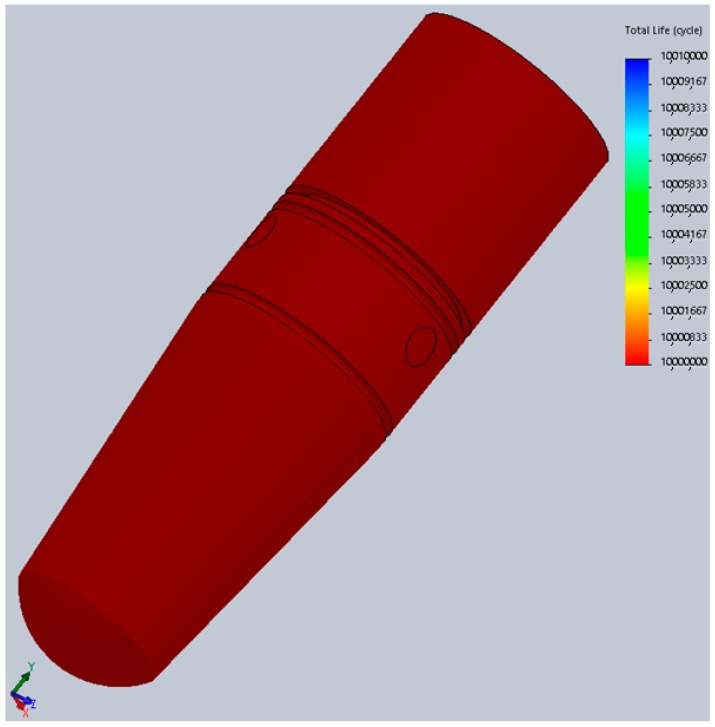
Life expectancy of tested implant prototype (FEA).

**Figure 16 materials-12-03444-f016:**
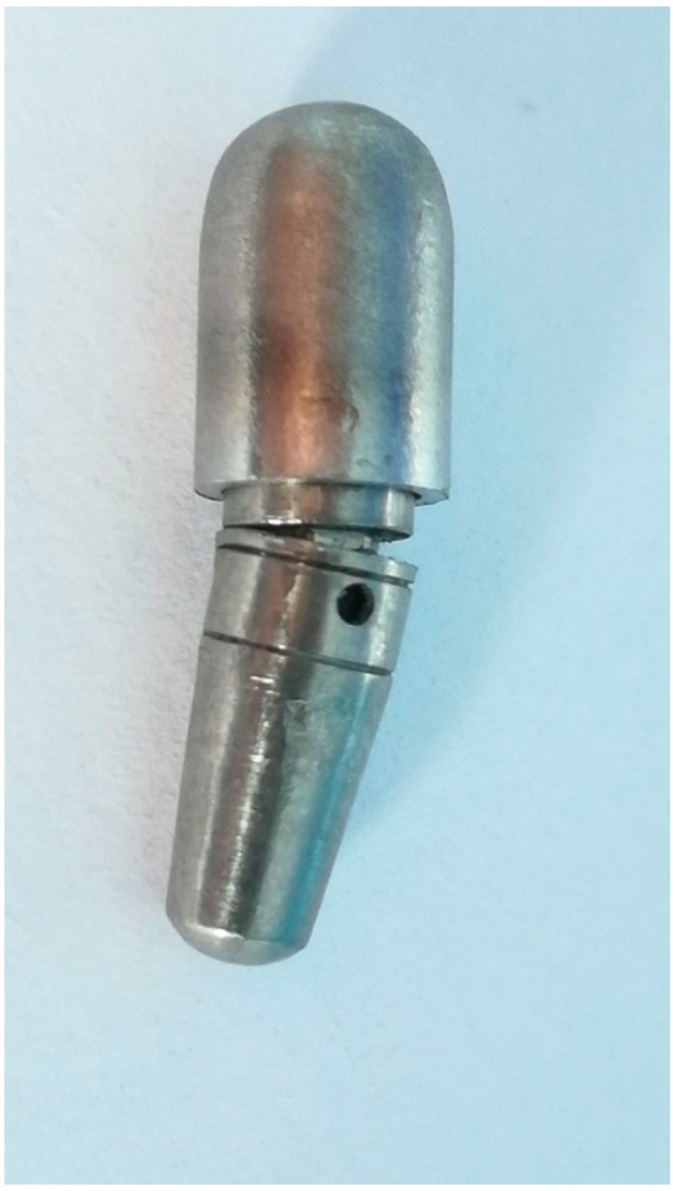
Failure of second tested implant due to test machine-related problems.

**Figure 17 materials-12-03444-f017:**
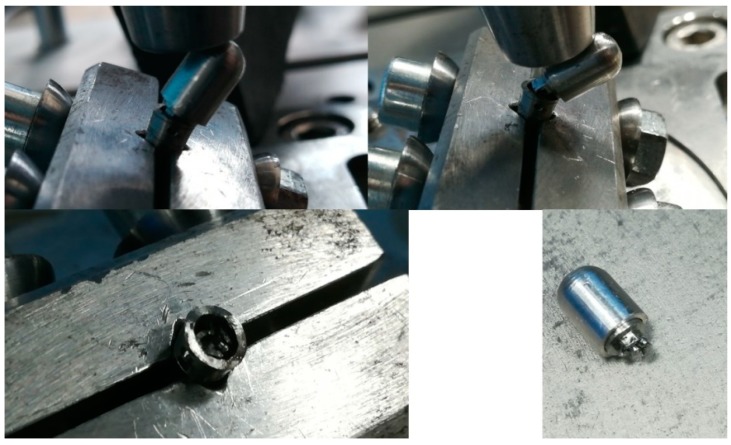
Failure of third implant during fatigue testing.

**Figure 18 materials-12-03444-f018:**
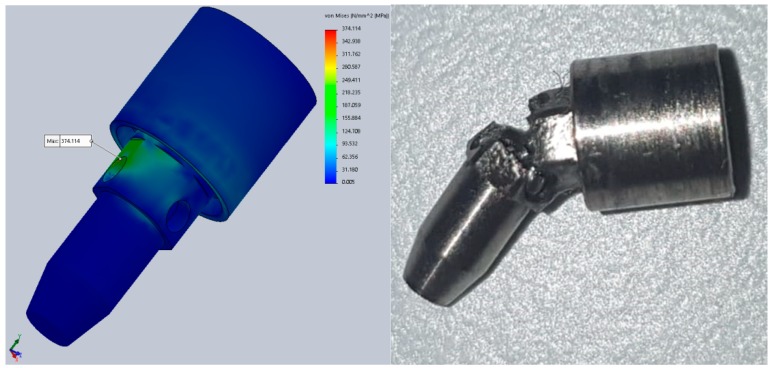
Comparison between FEA and physical fatigue tests.

**Table 1 materials-12-03444-t001:** Number of testing cycles for S1, P2, and P3.

No.	Cycle No.	Sample Shift (mm)			Observations
		S1	P2	P3	-
1	100	0.11485	0.21088	0.19126	-
5	500	0.11388	0.16948	0.16641	-
10	1000	0.11565	0.17079	0.16562	-
15	1500	0.11814	0.19228	0.16759	-
16	2000	0.11799	0.19170	0.16585	-
19	5000	0.11714	0.18998	0.16583	-
24	10,000	0.11628	0.19255	0.16246	Sample P2 broke
34	20,000	0.11471	-	0.20694	-
35	30,000	0.11270	-	0.21676	-
36	35046	-	-	0.21600	Sample P3 broke

**Table 2 materials-12-03444-t002:** Static testing for P1 and S2 implants.

Crt. No.	Force (N)	Displacement (mm)		Rigidity (N/mm)	
		P1	S2	P1	S2
1	50	0.219	0.024	228.3	2083.3
2	100	0.375	0.052	266.7	1923.1
3	150	0.500	0.082	300.0	1829.3
4	200	0.696	0.112	287.4	1785.7
5	250	0.845	0.136	295.9	1838.2
6	300	1.001	0.170	299.7	1764.7
7	350	1.107	0.200	316.2	1750.0
8	400	1.166	0.232	343.1	1724.1
9	450	1.224	0.258	367.6	1744.2
10	500	1.285	0.288	389.1	1736.1
11	548	1.347	0.318	406.8	1723.3
12	550	1.538	0.320	357.6	1718.8
13	600	1.646	0.350	364.5	1714.3
14	1400	-	1.620	-	864.2
15	1800	-	2.120	-	849.1
16	2177.5	-	3.048	-	714.4

**Table 3 materials-12-03444-t003:** The influence of implant placement (rotation) on the maximum stress.

α (°)	σ_ech,max_ (MPa)	α (°)	σ_ech,max_ (MPa)
0	349.6	190	298.5
10	361.6	200	300.4
20	371.2	210	314.1
30	374.1	220	323.1
40	370.3	230	327.1
50	362	240	327.4
60	355.4	250	332.5
70	342.7	260	334.8
80	330.3	270	325.3
90	341.5	280	316.3
100	347.5	290	311.7
110	348.7	300	316.6
120	344.9	310	319.5
130	335.7	320	317.9
140	327.8	330	311.3
150	321.6	340	327.8
160	313.2	350	339.6
170	313.7	360	349.6
180	308.8	-	-

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
