# Peer review of "New Dental Implant with 3D Shock Absorbers and Tooth-Like Mobility—Prototype Development, Finite Element Analysis (FEA), and Mechanical Testing"

_materials, 2019, doi:10.3390/ma12203444_

Round 1

Reviewer 1 Report

The paper is very interesting because it studies engineering models that, although they cannot give certain results, are very close to reality.

The finite element method is widely used in dentistry and is also reliable for these mechanical tests.

If possible I would like to suggest to the authors to better explain the materials and methods section as different engineering tests are carried out which may be difficult to understand for non-engineering readers

I would like to suggest to the authors to insert the following bibliographic entries that are very relevant to this paper:

1) FEM analysis of dental implant-abutment interface overdenture components and parametric evaluation of Equator® and Locator® prosthodontics attachments DOI: 10.3390/ma12040592

2)FEM investigation of the stress distribution over mandibular bone due to screwed overdenture positioned on dental implants DOI: 10.3390/ma11091512

Author Response

Esteemed reviewer,

Thank you for your very kind and extremely valuable input! The required modifications were made, helping increase the clarity and scientific value of our paper.

The tests principles were better described through the entire article. Both the references were inserted in the “Discussions” section and the references updated accordingly.

Reviewer 2 Report

I commend the authors for their novel dental implant design with 3D shock absorbing mechanism. The manuscript is thorough. Introduction was well written. All the materials used and methods followed were thoroughly explained. Figures and tables were neat and well represented. Conclusions drawn from the results were logical.

The only major issue I could see in the manuscript is the lack of testing of these implants in wet conditions. Since the application of these implants are in the mouth, a continuous wetting of the implant is unavoidable. I would strongly recommend quickly performing the tests in wet conditions and include them in the manuscript. If there is a strong justification of not performing wet studies, atleast include it in the manuscript. Other than that, I would recommend publishing the manuscript in the current stage. 

Author Response

Esteemed reviewer,

Thank you for your very kind and extremely valuable input! The required modifications were made, helping increase the clarity and scientific value of our paper.

The rationale for not conducting the test in a wet environment was mentioned in the “Material and Method” section. Unfortunately, since these tests take a very long time to perform (several days/implant) it would take a very long time before the testing machine would be available to us and another considerable period of time until our tests would be ready and their results interpreted. Another inconvenience of testing the implants in a wet environment as described by the ISO standard is the quality of the liquid present around the implant, since in the in vitro test it is fully immersed in saline solution which does not have the exact same composition as human saliva and is also present in an immense quantity around the implant when compared to the quantity of saliva present in the oral cavity at any point. The following stages of our tests will include such test though, to observe any potential differences between the results in the dry and wet environment but also to assess the differences in microleakage which should be considerably reduced in our implant compared to standard implant systems.

Reviewer 3 Report

The paper presents a new and original implant design which replicates the 3D physiological mobility of natural teeth.This paper seems to align with the journal's scope. However, the approach and results present in this paper is decent and can be improved. For example, further FEM analysis needs to be performed. This include, mesh refinement studies and sensitivity analysis of material parameters. If these are considered in the revised paper, then the paper would be strong and is suitable to be published in Material's journal.

Author Response

Esteemed reviewer,

Thank you for your very kind and extremely valuable input! We are honored to have our article reviewed by somebody with your expertise in the field of FEM Analysis. 

Your input was taken into account and mentioned in the article, helping increase the clarity and scientific value of our paper. Although unfortunately we did not get the chance to repeat the tests after mesh refinement and especially sensitivity analysis of material parameters due to the lack of time, they will be performed as soon as possible. These studies were planned for our future tests but we had a meeting with our colleagues from the Technical University yesterday to actually discuss the test parameters and start them sooner than initially planned.

Reviewer 4 Report

In my humble opinion, the article starts from some wrong conceptual assimilations that denote little knowledge of biology and histology as well as the concept of osseointegration as it is known until now.
To mention the most serious aspects:
1. Imagining to provide the mobility of a tooth to an implant means stressing the bone tissue that has achieved osseointegration with consequent loss of stability;
2. The lack of periodontal ligament (given by the ankylotic union between implant fixture and bone) cannot be replaced except by a ligamentous structure;
3. The authors believe that it is necessary to provide dental implants with the mobility of natural teeth to be able to make prosthetic mixed rehabilitations which (according to them) would not be mechanically feasible up to now. They should have read most recent scientific reports to learn that the connection between natural teeth and implants is now an accepted therapeutic option with a low incidence of biological and / or prosthetic complications.

Regards

Author Response

Esteemed reviewer,

Thank you for your kind and extremely valuable input! The required modifications were made, helping increase the clarity and scientific value of our paper.

The implant is not moving in relation to the surrounding bone which would indeed compromise the osseointegration and lead to bone resorption (as in the case of implants presenting periimplantitis). The clarification was inserted in the phrase on line 175-177: “The cushioning mechanism is placed between the implant and abutment. This is the only positioning that can provide the required amortization and mobility for the superstructure, without interfering with the exterior design of the implant and its osseointegration.” In this situation, the stress is absorbed mostly by the cushioning system and by the interior aspect of the implant body, decreasing the force transmitted to the periimplant tissues. The cushioning system inside the implant can be adapted in the following phases to better imitate a ligamentous structure if deemed necessary by future tests. Line 472-475: “The cushioning mechanism did not sustain any macroscopic damage but its presumed efficiency must be further tested. Also, its design can be modified in order to provide a more similar function to the ligamentous structure of the periodontal structures.” Although the concept of splinting dental implants with natural teeth is in some cases considered acceptable, it is still far from being considered the best treatment option. Many practicians avoid it basing their choice on published literature and personal clinical experience. The entire parahraph Line 444-459 was modified, inserting reference “42”, an extensive review of the subject.

The value of this article is not only limited to this aspect. It also describes the first steps of developing a new implant concept and several discoveries made during this process. Also, this is not the only implant feature.

Round 2

Reviewer 3 Report

The authors have provided satisfactory responses to my comments. I recommend publication.

Author Response

Esteemed reviewer, 

Thank you for your valuable input throughout this submission process. We will keep your observations in mind to guide us in our research

Reviewer 4 Report

Dear authors,

I recommended rejection since, in my opinion, there would not be possibility to ameliorate your work since it suffers from too many methodological points.

I confirm my previous decision.

Author Response

Esteemed reviewer,

Thank you for your valuable input.

We are sorry you feel this way and we are also willing to make any required modifications you see fit to improve the quality of our paper.